# A Photoluminescent Colorimetric Probe of Bovine Serum Albumin-Stabilized Gold Nanoclusters for New Psychoactive Substances: Cathinone Drugs in Seized Street Samples

**DOI:** 10.3390/s19163554

**Published:** 2019-08-15

**Authors:** Yao-Te Yen, Ting-Yueh Chen, Chun-Yu Chen, Chi-Lun Chang, San-Chong Chyueh, Huan-Tsung Chang

**Affiliations:** 1Department of Forensic Science, Ministry of Justice Investigation Bureau, Xindian District, New Taipei City 23149, Taiwan; 2Department of Chemistry, National Taiwan University, Taipei 10617, Taiwan; 3Department of Chemistry, National Cheng-Kung University, Tainan 70101, Taiwan

**Keywords:** gold nanoclusters, sensor, new psychoactive substances, cathinone

## Abstract

Screening of illicit drugs for new psychoactive substances—namely cathinone—at crime scenes is in high demand. A dual-emission bovine serum albumin-stabilized gold nanoclusters probe was synthesized and used for quantitation and screening of 4-chloromethcathinone and cathinone analogues in an aqueous solution. The photoluminescent (PL) color of the bovine serum albumin-stabilized Au nanoclusters (BSA-Au NCs) probe solution changed from red to dark blue during the identification of cathinone drugs when excited using a portable ultraviolet light-emitting diodes lamp (365 nm). This probe solution allows the PL color-changing point and limit of detection down to 10.0 and 0.14 mM, respectively, for 4-chloromethcathinone. The phenomenon of PL color-changing of BSA-Au NCs was attributed to its PL band at 650 nm, quenching through an electron transfer mechanism. The probe solution was highly specific to cathinone drugs, over other popular illicit drugs, including heroin, cocaine, ketamine, and methamphetamine. The practicality of this BSA-Au NCs probe was assessed by using it to screen illicit drugs seized by law enforcement officers. All 20 actual cases from street and smuggling samples were validated using this BSA-Au NCs probe solution and then confirmed using gas chromatography–mass spectrometry. The results reveal this BSA-Au NCs probe solution is practical for screening cathinone drugs at crime scenes.

## 1. Introduction

Synthesized cathinones, a type of new psychoactive substances (NPS), have emerged in the past 10 years and have caused serious social problems worldwide [1]. Cathinone drug abuse has adverse psychiatric effects, including anxiety, anorexia, addiction, and panic attacks, leading to homicidal combative behavior and deaths, which puts a community in danger [2]. More than 30 synthesized cathinone analogues derived from natural cathinone with main structure of phenyl-ketone group had been reported by 2018. The fast appearance of cathinone drugs on the drug market poses a considerable challenge for law enforcement officers and analytical chemists [3].

Cathinone drugs in street samples are usually present in a powder form and common for drug abusers to use. The concentrations of illicit substances in these samples are dependent on the sources sold from drug dealers. Before seizing these illicit drug samples, law enforcement officers need to identify substances by using sensor kits or instruments. Although gas chromatography–mass spectrometry (GC–MS) and liquid chromatography–tandem MS (LC–MS/MS) methods for quantitation of cathinone drugs and their metabolites have been developed [4,5,6,7], these instruments were difficult to perform at crime scenes. To overcome this obstacle, a portable mass spectrometer equipped with desorption electrospray ionization used at crime scenes was developed for the direct analysis of cathinone drugs with limits of detection (LODs) down to the nano-gram level [8]. In addition, a handheld Raman spectrometry has been developed and is available commercially with an advantage for in-field testing [9]. Besides, a few colorimetric sensors and electrochemical probes as well as immunoassays to detect cathinone drugs have been developed for screening drugs at crime scenes [10,11,12]. Cuypers et al. used classic color testing arrays (Marquis, Mecke, Scott, Mandelin, and Simon reagents) to identify NPS [10]. Banks et al. used graphite-screen-printed electrodes for quantification of mephedrone and 4′-methyl-N-ethylcathinone with their limits of detection down to 11.80 and 11.60 μg mL^−1^, respectively [11]. The Randox Drugs of Abuse V Biochip is a commercially available immunoassay for cathinone drugs detection [13]. All aforementioned methods have some shortcomings regarding their performance for screening drugs at crime scenes, such as flowchart complications, the need for expertise, cost inefficiency, and unpopularity. Thus, there is a high demand to develop a sensitive, low-cost, easy-to-use, and high-specificity sensor for cathinone drugs screening.

Nanoclusters (NCs) have attracted a lot of attention in nanotechnology because of their fantastic chemical and physical properties [14,15]. Many scientists have attempted to synthesize photoluminescent (PL) metallic NCs using different processes, including etching [16], sonochemical [17], and template-based methods [18]. Currently, metallic NCs such as Au, Ag, Cu, and AuAg NCs have been synthesized [19,20,21,22]. Owing to its photostability and facile synthetic properties, the bovine serum albumin-stabilized Au NCs (BSA-Au NCs) is relatively popular for metallic NCs. BSA-Au NCs composed of several gold atoms at the core and BSA at the outer-layer surface have applications in cancer therapy, bioimaging, and sensors [23,24,25]. With an aim to solve a shortage problem for cathinone drug sensors, we report a PL colorimetric probe of BSA-Au NCs changed from red to dark blue, visible to the naked eye, during the identification of cathinone drugs. We demonstrated the facile synthetic property, high sensitivity, and excellent selectivity of BSA-Au NCs for quantitation of 4-chloromethcathinone (4-CMC) in an aqueous solution. The sensing mechanism of BSA-Au NCs for quantitation of 4-CMC was proposed regarding a Stern-Volmer plot, absorption, PL, and lifetime data. In addition, 20 actual cases acquired from street and smuggling samples were validated using a BSA-Au NCs probe solution. The results revealed that the use of a BSA-Au NCs probe to detect cathinone drugs at crime scenes was practical.

## 2. Materials and Methods

### 2.1. Reagents and Materials

BSA, fructose, glucose, hydrogen tetrachloroaurate (III) trihydrate (HAuCl_4_·3H_2_O), quinine sulfate monohydrate, sodium hydroxide (NaOH), sodium borohydride (NaBH_4_), and sucrose were purchased from Sigma-Aldrich (St. Louis, MO, USA). Monobasic and dibasic sodium phosphates were purchased from J. T. Baker (Phillipsburg, NJ, USA) and thus were used to prepare sodium phosphate buffers (100 mM, pH 7.0). A Milli-Q system from Millipore (Billerica, MA, USA) was used in this study to produce ultrapure water (18.2 MΩ cm). Butylone, 4′-chloro-α-pyrrolidinopropiophenone (4′-chloro-α-PPP), 4-CMC, cocaine, dibutylone, ephylone, ethylone, heroin, ketamine, 3-methoxymethcathinone (3-MeOMC), 4-methyl-α-ethylaminopentiophenone (4-MEAPP), methamphetamine, mexedrone, and pentylone in hydrochloride salt form were acquired from actual cases.

### 2.2. Synthesis of BSA-Au NCs

The synthetic process in this study was similar to that described in relevant research, with moderate modifications [26]. An aliquot (1.0 mL) of HAuCl_4 (aq)_ (10 mM) was mixed with 10 mL of BSA _(aq)_ (30 mg mL^−1^) in a 50-mL centrifugation tube. After stirring for 30 min, 0.5 mL of NaOH _(aq)_ (1 M) was added to the mixture and incubated at 45 °C for 18 h. The color of the solution changed from light yellow to dark yellow, indicating the formation of BSA-Au NCs. With the aim to remove unreactive gold species and to purify the resulting BSA-Au NCs solution, the mixture was purified against ultrapure water through a dialysis membrane (MWCO 3.5 kD) purchased from Spectra Labs (Rancho Dominguez, CA, USA) for 24 h. Next, the purified BSA-Au NCs solution was freeze-dried to form a light pink powder (259 mg), which was then re-dispersed in sodium phosphate buffers (pH 7.0, 100 mM). The as-prepared BSA-Au NCs solution (22.5 mg mL^−1^) was stored at 4 °C in the dark until ready to use.

### 2.3. Characterization of BSA-Au NCs

The BSA-Au NCs light pink powder (0.1 mg) was diluted with ultrapure water (1 mL). The diluted BSA-Au NCs were deposited on 400-mesh carbon-coated Cu grids, and excess solvent was evaporated at an ambient temperature and pressure. Images of BSA-Au NCs were captured using transmission electron microscopy (TEM) purchased from GCE Market (FEI Tecnai-G2-F20, Blackwood, NJ, USA) operated at 200 kV to measure particle sizes. A SpectraMax i3x Multi-Mode Microplate Reader from Molecular Devices (San Jose, CA, USA) was used to record the absorption spectrum and PL spectrum of BSA-Au NCs. A spectrometer (Theta Probe) from VG Scientific (East Grinstead, UK) with Al Kα X-ray radiation was used to measure X-ray photoelectron spectroscopy (XPS) of BSA-Au NCs. A Thermo Scientific spectrometer (Nicolet iS5, Waltham, MA, USA) with a resolution of 4 cm^−1^ and 64 scans was used to record the Fourier transform infrared (FTIR) spectrum of BSA-Au NCs and BSA to analyze the functional groups on the surface. Fluorescent quantum yields (Φ_f_) of the BSA-Au NCs dispersed in a sodium phosphate buffer (pH 7.0, 100 mM) at the excitation and emission wavelengths of 365/460 nm and 365/650 nm, were separately measured using quinine sulfate (Φ_f_ = 0.54) dissolved in 0.1 M H_2_SO_4_ as a standard [27].

### 2.4. Detection of 4-CMC and Cathinone Analogues

An aliquot (100 μL) of the as-prepared BSA-Au NCs solution was dispersed in a sodium phosphate buffer (400 μL, 100 mM, pH 7.0) containing various concentrations of 4-CMC to evaluate the sensitivity of BSA-Au NCs for detection of 4-CMC. After mixing for 1 min, the PL spectrum of the BSA-Au NCs solution with and without 4-CMC were recorded. Similarly, a selectivity test was assessed using 25 mM of other illicit drugs, including heroin, cocaine, methamphetamine, and ketamine dissolved in sodium phosphate buffers, separately. Additives such as glucose, sucrose, and fructose were tested for their interference. In addition, cathinone analogues, including butylone, 4′-chloro-α-PPP, dibutylone, ephylone, ethylone, 3-MeOMC, 4-MEAPP, mexedrone, and pentylone were tested to confirm the sensing ability of BSA-Au NCs probe for detection of cathinone drugs. All experiments were performed under ambient conditions (25 °C, 1 atm). To investigate the sensing mechanism, a reduction of 4-CMC to form 1-(4-chlorophenyl)-2-(methylamino) propan-1-ol was synthesized by using a reducing reagent of NaBH_4_ (Appendix A) to study analyte-induced PL quenching. In addition, time-resolved absorption and emission spectroscopy (NS010, Pascher Instrument, Lund, Sweden) was performed using a photon-counting system with a 365 nm laser to measure lifetimes of BSA alone and BSA-Au NCs with and without 4-CMC (25 mM), which were dispersed in sodium phosphate buffers (100 mM, pH 7.0).

### 2.5. Interference Study

An aliquot (100 μL) of the as-prepared BSA-Au NCs solution was mixed with 400 μL of the sodium phosphate buffer (100 mM, pH 7.0) in a 2-mL centrifugation tube for use as a BSA-Au NCs probe solution. Various concentrations of 4-CMC mixed with glucose (*w*/*w*% from 0% to 20%) were individually prepared for interference study. Each real sample (30 mg) was dissolved in 500 μL of a BSA-Au NCs probe solution. After being mixed for 1 min, PL images of the BSA-Au NCs probe solution were recorded using a smartphone (Galaxy Note 3, Samsung, Seoul, South Korea) and a portable UV–LED lamp (FLUV Series, LEDWELL, New Taipei City, Taiwan) at an excitation wavelength of 365 nm.

### 2.6. Theoretical Calculation

The structures of 4-CMC and its reduction form of 1-(4-chlorophenyl)-2-(methylamino) propan-1-ol were optimized by using Gaussian 09 [28]. Then, density-functional theory (DFT) calculation based on B3LYP/6-31G** mode was used to calculate electron distribution and energy levels of the two analytes for their highest occupied molecular orbitals (HOMOs) and lowest unoccupied molecular orbitals (LUMOs). Molecular orbitals were displayed using a Chemcrafe.

### 2.7. Analysis of Actual Samples

Twenty actual cases from street and smuggling samples from different sources were validated by using a BSA-Au NCs probe solution and a portable UV–LED lamp, and then they were confirmed using GC–MS measurement (Agilent 7890A/5975C MSD). The procedure was similar to that used for interference study, but only 2 to 5 mg of unknown samples were sampled for screening.

## 3. Results and Discussion

### 3.1. Characterization of BSA-Au NCs

The TEM image displayed in Figure 1A shows that the as-prepared BSA-Au NCs are uniform and monodispersed spheres, with an average diameter of 3.2 nm (100 counts) [29]. The XPS spectrum presented in Figure 1B reveals the existence of various elements in BSA-Au NCs, including Au, O, N, and C. Binding energies of 285.1, 399.1, and 531.1 eV were attributed to C_1s_, N_1s_, and O_1s_, respectively [30]. In the case of Au, binding energies of 84.3 and 87.6 eV were attributed to 4f_7/2_ and 4f_5/2_, respectively, and this further confirmed the formation of BSA-Au NCs [30]. Moreover, the deconvoluted Au XPS spectrum of BSA-Au NCs revealed four peaks obtained at 84.3 and 87.8, 85.7, and 89.6 eV, which were attributed to oxidation states of Au (0) and Au (I), respectively, as shown in Figure 1C [31,32]. The number of gold atoms in the BSA-Au NCs was proposed approximately to be 25, due to its red emission from Au clusters [32,33,34]. As indicated in Appendix A, the FTIR spectrum of BSA-Au NCs was similar to that of BSA, further confirming that Au NCs was stabilized in the BSA [30]. The absorption and PL spectrum of the BSA and BSA-Au NCs are provided in Figure 1D. The BSA-Au NCs had a significant absorption band at a wavelength of 280 nm that was attributable to the tryptophan in BSA [35]. The absorption spectrum of BSA-Au NCs was similar to that of BSA alone, further indicating that BSA was stabilized on the surface of BSA-Au NCs. The PL spectrum of BSA-Au NCs showed two emission bands at 460 nm and 650 nm when excited at 365 nm. The emission bands of BSA-Au NCs at 460 nm and 650 nm were blue and red in color and corresponded to BSA and Au NCs, respectively. Table 1 summarizes the lifetimes of BSA and BSA-Au NCs at PL bands of 460 nm and 650 nm at an excitation wavelength of 365 nm. The lifetimes of BSA-Au NCs at PL bands of 460 nm and 650 nm were 10.5 ns and 1820.7 ns, respectively [36], and the lifetime of BSA at the PL band of 460 nm was 11.3 ns, indicating that the PL band at 460 nm of BSA-Au NCs should be attributed to BSA. Fluorescent quantum yield values for BSA-Au NCs at 460 nm and 650 nm were 1.2% and 1.5%, respectively. The stability of BSA-Au NCs against time and pH was assessed as shown in Appendix A. The results revealed that the PL intensities of BSA-Au NCs for a time of 480 min and pH values in the range of 7–11.5 were stable, demonstrating excellent stability for use. After pH values (7, 9.5, and 11.5) was assessed, a mild pH of 7.0 was selected for further study [37].

### 3.2. Sensitivity, Selectivity, and Interference Study

Figure 2A displays the PL spectrum of BSA-Au NCs in a sodium phosphate buffer (pH 7.0) with and without 4-CMC when excited at 365 nm. The analyte-induced PL band of the BSA-Au NCs at 650 nm exhibited an obvious decrease with increasing concentrations. Figure 2A (inset) shows that the relationship (F_0_−F)/F_0_ of the BSA-Au NCs is linear for the range of 0.48–7.5 mM of 4-CMC (R^2^ = 0.998), where F and F_0_ are the PL intensities of the BSA-Au NCs at 650 nm with and without 4-CMC, respectively. The LOD was calculated to be 0.14 mM based on 3δ/slope, where δ represented the deviation of (F_0_−F)/F_0_ for a blank test and the slope denoted the calibration curve. As shown in Figure 2B, the results of the selectivity test demonstrate that the BSA-Au NCs probe is specific to 4-CMC and cathinone analogues (their structures are displayed in Appendix A). In addition, common illicit drugs and additives, including cocaine, heroin, ketamine, methamphetamine, glucose, sucrose, and fructose, did not induce BSA-Au NCs quenching. Figure 3A presents the PL color of the BSA-Au NCs probe solutions with increasing concentrations of 4-CMC. The change in PL color from red to dark blue was visible to the naked eye; thus, it was determined that the changing point was 10.0 mM, indicating that a small amount of the analyte (1 mg) was enough for this probe solution to change PL color. Figure 3B shows that the 4-CMC-induced PL color of BSA-Au NCs probe solution changed mixing with glucose at various concentration levels (*w*/*w*% from 0% to 25%). When the concentration of 4-CMC was higher than 5%, the PL color of BSA-Au NCs was completely blue, indicating high concentration of glucose interference would not change the sensing results of BSA-Au NCs for the screening of 4-CMC.

### 3.3. Sensing Mechanism

A Stern–Volmer plot, as displayed in Figure 4A, was used to study the quenching mechanism of BSA-Au NCs for 4-CMC at emission and excitation wavelengths of 650 nm and 365 nm. The slope (K_SV_) of the linear plot was determined to be 0.269 (R^2^ = 0.996), revealing a dynamic quenching mechanism (fluorescence resonance energy transfer or electron transfer) [38]. In addition, the lifetimes of BSA-Au NCs without and with 25 mM of 4-CMC at emission and excitation wavelengths of 650 nm and 365 nm were determined to be 1820.7 and 963.4 ns, respectively [39]. To confirm fluorescence resonance energy transfer or electron transfer, the PL spectrum of BSA-Au NCs at an excitation wavelength of 365 nm and the absorption spectrum of 4-CMC were plotted, as displayed in Figure 4B. The results revealed no overlap between the PL spectrum of BSA-Au NCs in the range of 530–730 nm and the absorption spectrum of 4-CMC, ruling out fluorescence resonance energy transfer [40,41]. Thus, an electron transfer occurred from BSA-Au NCs to 4-CMC when excited at 365 nm, mainly due to the phenyl-ketone structure in 4-CMC, which was a good electron acceptor [42]. To prove this, 4-CMC was reduced to form 1-(4-chlorophenyl)-2-(methylamino) propan-1-ol using NaBH_4_ as shown in Appendix A; thus, the analyte did not induce PL quenching of BSA-Au NCs. Also, electron distribution and energy levels of HOMO and LUMO for 4-CMC were acquired by DFT calculation. As shown in Figure 4C, the HOMO orbitals are localized on the amino group, and the LUMO orbitals are delocalized on a phenyl-ketone structure. Their HOMO and LUMO energy levels are –5.90 and –1.83 eV, respectively. The DFT calculation results further support the phenyl-ketone group in 4-CMC, which would accept an electron from BSA-Au NCs. On the other hand, HOMO and LUMO energy levels of 1-(4-chlorophenyl)-2-(methylamino) propan-1-ol are–6.51 and –0.43 eV, respectively, indicating the high energy level of LUMO would not allow an electron transfer from BSA-Au NCs. On the basis of the Stern–Volmer plot, the lifetimes, absorption, PL, and experiment results of a reductive form of 4-CMC as well as DFT calculation data, a sensing mechanism of BSA-Au NCs for 4-CMC, is proposed, as displayed in Scheme 1.

### 3.4. Analysis of Actual Samples

Twenty actual cases from street and smuggling samples were validated by using this BSA-Au NCs probe solution and then confirmed using GC–MS measurements, as shown in Figure 5. Actual cases of 2, 3, 11, and 16 screened positive results and were thus confirmed to be ephylone, butylone, a mixture of dibutylone and 4-chloro-*N*,*N*-dimethylcathinone (4-chloro-*N*,*N*-DMC), and 4-chloro-*N*,*N*-DMC, respectively, which all were cathinone drugs. Other cases tested negative and were thus confirmed to be amphetamine, ketamine, methoxetamine, cocaine, 2,5-dimethoxy-4-bromophenethylamine (2C-B), 3,4-methylenedioxymethamphetamine (MDMA), ACHMINACA, 5-chloro-AKB48, fentanyl, etizolam, and heroin, respectively. All screening results fit the outcomes confirmed by GC–MS measurements. Moreover, it was noted that PL images of BSA-Au NCs probe solutions were captured using a smartphone and a portable UV–LED lamp, indicating that the developed sensor system was practical and convenient for law enforcement officers to use at crime scenes.

## 4. Conclusions

A PL color-changing probe of BSA-Au NCs has been developed for screening and quantitating 4-CMC. This probe is specific to cathinone drugs, including butylone, 4′-chloro-α-PPP, 4-CMC dibutylone, ephylone, ethylone, 4-MEAPP, 3-MeOMC, mexedrone, and pentylone. When excited at 365 nm, analyte-induced PL quenching of BSA-Au NCs at 650 nm occurs through an electron transfer, causing the PL color to change from red to dark blue. Using a smartphone and a portable UV–LED lamp, this simple and easy-to-use probe of BSA-Au NCs will enable law enforcement officers to rapidly screen cathinone drugs at crime scenes with the naked eye.

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
