# Peer review of "A Photoluminescent Colorimetric Probe of Bovine Serum Albumin-Stabilized Gold Nanoclusters for New Psychoactive Substances: Cathinone Drugs in Seized Street Samples"

_sensors, 2019, doi:10.3390/s19163554_

Round 1
Reviewer 1 Report
In this article, the authors present a novel application of using luminescent BSA-Au NCs for quantitation and screening of 4-chloromethacathinone and other cathinone-drugs in an aqueous solution. While this is a relatively comprehensive study, the following concerns should be addressed:
Line 54, this comment is not accurate. Not all NC have fantastic biocompatibilities and photophysical properties. Not even gold nanoclusters. Line 116, The authors detected the PL of the mixed solution in 1 min. Will the PL of the mixed solution further degrade with time (after 10min, 1h, or even 24hr)? In Figure 1b, the zoomed-in XPS figure seems to have more than two peaks. What’s the charge of Au here (0 or +1)? Line 160, the authors’ statement is not accurate. The m/z of the Au NCs ranges from ~64000 to ~69000. For a larger m/z, there are certainly more than 7 Au atoms. Line 162, the conclusion here is misunderstanding. “BSA was stabilized on the surface of BSA-Au NCs”. FTIR figure does not show interaction between BSA and Au NCs. Also, this statement is indicating that there are multiple layers of BSA on the surface of Au NCs. Is there free BSA in the solution? The purification method reported here can only remove unreacted gold species. The emission peak of BSA is 350nm in previous reports. I understand that the authors used 365nm excitation. Are they only monitoring the tail of BSA PL? Although high glucose concentration won’t change the result that Au NCs PL be quenched by 4-chloromethacathinone, does the sensitivity match the work function in Figure 2A? Is 4-chloromethacathinone conjugated to BSA-Au? Is the PL quenching reversible?Overall, I would recommend the publication of this article in Sensors if the above concerns are solved.
Reviewer 2 Report
Revision regarding manuscript ID 574032 entitled a photoluminescent colorimetric probe of bovine serum albumin-stabilised gold nanocluster for new psychoactive substances: cathinone-drugs in seized street samples
The manuscript addresses an important topic and communicates interesting results but still needs some improvement.
The introduction should be contextualised more to the topic of the manuscript.
The grammar and language needs revising.
Be consistent in the use of tense and do not mix past with present.
The abstract gave a good overview on the research and communicated the results clearly but it still needs improvement in the structure. For instant state the aim before explaining the method. Remove the part regarding confirmation with GCMS. This could be added in the experimental section of the main text not in the abstract section.
Add the abbreviation between brackets where first mentioned then use it throughout the text.
Line 34: give examples of the social problems associated with cathinones
Line 36: reference 2 does not support what is stated.
Use more updated citations from the united nations and EMCDDA
Lines 38-40: I would suggest removing the pharmacology as this paper is more concerned with identifying street samples. Explain the context of cathinones on the streets, what derivatives are present and why there is a need to identify them? Then discuss the relevant literature for their analysis and justify the need for this method.
The justifications in lines 44-53 is weak. The methods should be presented in details.
There is a vast majority of studies regarding rapid and portable techniques for identifying cathinones. These studies seem to be ignored.
Line 63: what other cathinones were characterised?
Lines 81-82: Move to method section rather than material section.
Lines 77-78: Arrange solvents alphabetically and do this for all lists.
Line 105: What were the instrumental settings used? Explain
Line 107: Same as 105, specify the settings
Line 147: Was the composition of the street sample similar? Were they from the same source?
Line 231 :Is the drug ethylone or ephylone?
Add the street names of the drugs in the appendix not just the cathinones
